# Study on Spatiotemporal Variation Pattern of Vegetation Coverage on Qinghai–Tibet Plateau and the Analysis of Its Climate Driving Factors

**DOI:** 10.3390/ijerph19148836

**Published:** 2022-07-21

**Authors:** Xiaoyu Deng, Liangxu Wu, Chengjin He, Huaiyong Shao

**Affiliations:** 1College of Earth Sciences, Chengdu University of Technology, Chengdu 610059, China; dxy@cdut.edu.cn (X.D.); 2020020014@stu.cdut.edu.cn (C.H.); 2College of Global Change and Earth System Science, Beijing Normal University, Beijing 100875, China; 202121490037@mail.bnu.edu.cn

**Keywords:** vegetation coverage, ESTARFM, data fusion, climate change, topography

## Abstract

As one of the most sensitive areas to global environmental change, especially global climate change, the Qinghai–Tibet Plateau is an ideal area for studying global climate change and ecosystems. There are few studies on the analysis of the vegetation’s driving factors on the Qinghai–Tibet Plateau based on large-scale and high-resolution data due to the incompetence of satellite sensors. In order to study the long-term vegetation spatiotemporal pattern and its driving factors, this study used the enhanced spatial and temporal adaptive reflectance fusion model (ESTARFM) to improve the spatial resolution of the GIMMS NDVI3g (8 km) data of the Qinghai–Tibet Plateau in 1990 and 1995 based on the MODIS NDVI (500 m) data. The research on the spatiotemporal pattern and driving factors of vegetation on the Qinghai–Tibet Plateau from 1990 to 2015 was carried out afterward, with combined data including topographic factors, annual average temperature, and annual precipitation. The results showed that there was a strong correlation between the actual MODIS NDVI image and the fused GIMMS NDVI3g image, which means that the accuracy of the fused GIMMS NDVI3g image is reliable and can provide basic data for the accurate evaluation of the spatial and temporal patterns of vegetation on the Qinghai–Tibet Plateau. From 1990 to 2015, the overall vegetation coverage of the Qinghai–Tibet Plateau showed a degrading trend at a rate of −0.41%, and the degradation trend of vegetation coverage was the weakest when the slope was ≥25°. Due to the influence of the policy of returning farmland to forests, the overall degradation trend has gradually weakened. The significant changes in vegetation in 2010 can be attributed to the difference in the spatial distribution of climatic factors such as temperature and precipitation. The area with reduced vegetation in the west was larger than the area with increased vegetation in the east. The effects of temperature and precipitation on the distribution, direction, and degradation level of vegetation coverage were varied by the areal differentiation in different zones.

## 1. Introduction

Vegetation plays an irreplaceable role in the global mass–energy exchange, and it is interactive with climate change by affecting water, the carbon cycle, and soil composition [1,2,3]. Some studies have linked the vegetation change on the Qinghai–Tibet Plateau to global warming [4]. The Qinghai–Tibet Plateau, known as “ the third pole of the earth”, is located in a high-altitude and high-cold environment [5]. Its fragile ecosystem and complex geographic environment have resulted in a drastic change in the vegetation dynamics in the region and made it more sensitive to the change in global climate and ecosystems. Thus, it is considered to be an ideal research area for vegetation cover dynamics and driving force analysis research [6]. For example, Zhou et al. found that the vegetation on the Qinghai–Tibet Plateau exhibited an overall green profile from 1982 to 2012, using GIMMS NDVI3g data [7]. Xu et al. found that the vegetation coverage of the Qinghai–Tibet Plateau increased from 1982 to 2000 by extracting vegetation coverage derived information from the leaf area index satellite images [8]. Zhang et al. compared the results of vegetation trend changes based on vegetation indices from multiple data sources and found that different data sources may cause variance in trend changes [9]. Therefore, the dynamic monitoring of vegetation coverage on the Qinghai–Tibet Plateau and the analysis of driving factors are of great significance to the evaluation of construction in the Qinghai–Tibet Plateau ecological environment and environmental protection projects.

It should be noted that the current remote sensing data sources usually cannot have both a long time series and a high spatial resolution. For example, predecessors studied the vegetation changes in Northeast China from 1982 to 2009 [10], on the Mongolian Plateau from 1982 to 2006 [11], in Xinjiang from 1982 to 2013 [12], and on the Qinghai–Tibet Plateau from 1982 to 2015 [13] using GIMMS NDVI3g data, which is one of the longest time-series NDVI data sets [14]. However, the spatial resolution of this data was low, which was suitable for large-scale research, and had a weak ability to describe changes in regional small-scale vegetation coverage. The data collected by satellite sensors, such as the Earth Observation System (SPOT)-VEGETATION, MODIS, and ENVISAT Medium Resolution Imaging Spectrometer (MERIS), have a high time resolution. The effects of clouds can be reduced or even eliminated by synthesizing images within a certain period [14]. Thus, the quality of remote sensing data is improved, which expands the usage of the regional vegetation trend analysis on a global scale [15,16,17]. However, the demand for improving the spatial resolution of the data remains unsolved. Although in comparison with GIMMS NDVI3g, MODIS NDVI products have higher accuracy and can better express the time-series changes of vegetation. Many scholars have analyzed the changes in vegetation cover and its influencing factors on the Qinghai–Tibet Plateau with different temporal and spatial resolutions based on MODIS data [13,18,19]. It is inappropriate to be applied in studies of long time series due to its time coverage (July 2001 to present). Therefore, there is a need to improve the resolution of early remote sensing data for long-term vegetation change monitoring research.

There are great differences in vegetation types and distribution among different regions, so it is difficult to evaluate the regional vegetation growth status based on a single data set. It is urgent to reconstruct multi-source spatiotemporal data. At present, the spatiotemporal fusion methods based on remote sensing data were roughly divided into two types: on the one hand, the transformation model is based on systematic error, the main idea of which was that the reflectivity of different objects in different data sources has strong consistency, and the deviation was mainly caused by the system error of the sensor itself, so it was necessary to establish a certain regular conversion relationship for the data of different sensors. Li et al. established the data mapping relationship between pixels through pixel-by-pixel regression analysis and analyzed the response of vegetation dynamic changes on the Qinghai–Tibet Plateau to hydrothermal conditions in the past 40 years [20]. Zhang et al. obtained long-term remote sensing data by establishing a linear regression equation between GIMMS NDVI3g and MODIS data [21]. On the other hand, the principle of the data fusion method based on the reconstruction model was to realize the fusion of temporal and spatial information between sensors of different data sources by establishing the relationship between different data sources.

Predecessors have done a lot of research on the fusion of remote sensing data in time and space. For example, the STARFM (spatial and temporal reflection fusion model) proposed by Gao et al. [22] can fuse Landsat and MODIS surface reflectance to generate synthetic reflectance data with high temporal and spatial resolution. Using the STARFM model, Semmens et al. effectively monitored crop water use and soil water status at the farmland scale [23], while Singh fully extracted the phenology of the crop during the 8-year growing season [24]. Gevaert and Javier Garcia-Haro evaluated STARFM and believe that it is practically more effective in constructing time profiles with more high-resolution images [25]. Bhandari et al. [26] and Schmidt et al. [27] evaluated the ability of STARFM to synthesize long-term time-series data and obtained the phenological characteristics of different vegetation for research. Zhu et al. improved the STARFM algorithm and proposed the enhanced spatial and temporal adaptive reflectance fusion model (ESTARFM) to reflect the temporal and spatial changes of the land surface more accurately [28]. Moreover, in areas with complex terrain, ESTARFM is frequently used to generate synthetic remote sensing images with high spatial and temporal resolution due to its extraordinary fusion effect [29,30]. This method not only considered the temporal variation characteristics of pixel reflectivity but also considered the spatial reflectivity variation characteristics of pixel reflectivity, which effectively improved the accuracy of fusion. However, very few studies have focused on the synthesis of the GIMMS NDVI3g and MODIS NDVI on the Qinghai–Tibet Plateau spatiotemporally based on ESTARFM and further analyzed the spatiotemporal pattern of vegetation from 1990 to the present. For this reason, based on the ESTARFM, the present article developed spatiotemporal fusion of the Qinghai–Tibet Plateau, using GIMMS NDVI3g and MODIS NDVI to monitor the evolution of vegetation.

Therefore, this study used GIMMS NDVI3g and MODIS NDVI data to analyze the temporal and spatial patterns and driving factors of vegetation cover on the Qinghai–Tibet Plateau. Its main purposes were as follows. Firstly, we used the ESTARFM model to establish the functional relationship between the two data sources to improve the spatial resolution of the GIMMS NDVI3g data. Secondly, we quantified the vegetation coverage of the Qinghai–Tibet Plateau. Finally, we analyzed long-term vegetation change characteristics and their influencing factors on the Qinghai–Tibet Plateau.

## 2. Materials and Methods

### 2.1. Study Area

Known as the “roof of the world”, the Qinghai–Tibet Plateau (26°00′ N–39°47′ N, 73°19′ E–104°47′ E) is located in the mid-latitude alpine area with an average altitude of higher than 4000 m, an area of about 2.57 million km^2^, and solar radiation of 0.5861–0.7954 MJ cm^−1^ a^−1^ and is the largest area in China with the highest solar radiation value. This region has a complex terrain, numerous towering mountains, such as the Tanggula Mountains, Kunlun Mountains, and the Himalayas. The connection of the major mountains causes a huge terrain gap in the territory, showing an obvious altitudinal zonality. The Qinghai–Tibet Plateau is therefore characterized by high altitudes, forming natural conditions significantly different from other mid-latitude warm temperate zones and subtropical regions [31]. The weather is cold all year round with the average annual temperature in the hinterland below 0 °C. Moreover, the uneven distribution of precipitation, such as in the Himalayan mountains, greatly blocks the northward movement of the warm and wet air in the southern ocean, resulting in the decline of annual precipitation from 2000 mm to below 50 mm in the region and a climate type change from the humid southeast to the arid northwest [5]. The climate difference between different regions is obvious, the dry and wet were distinct, the northwest region is cold and arid, and the southeast region is warmer, hotter, and more humid. The Qinghai–Tibet Plateau has a long snow cover in winter and covers a large area, which is the source of the Yellow River, the Yangtze River, the Yarlung Zangbo River, and other important rivers. The Qinghai–Tibet Plateau contains rich herbaceous resources, and the main vegetation types are mountain forests, mountain shrubs, alpine meadows, alpine grasslands, alpine deserts, etc. [32]. Due to the harsh climate and high altitude, human activities are scarce. The local natural vegetation is well preserved. Such a unique geographical location and climate characteristics make the Tibetan Plateau a “sensor” of global climate change, where the vegetation is highly sensitive to climate change. As an eco-environmental fragile area, this region serves as an important ecological barrier in China. The vegetation changes pose a significant impact on the hydrological characteristics, carbon cycle, and surface stability of this area’s surface [33] as well as the climate and ecology of China or even the world. Therefore, it is of great significance to explore the spatial and temporal distribution characteristics of vegetation cover on the Qinghai–Tibet Plateau and analyze the main driving factors of regional vegetation changes.

### 2.2. Data Sources and Pre-Processing

#### 2.2.1. NDVI Data

As one of the sensitive indicators of vegetation changes, NDVI is widely applied in large-scale and regional vegetation monitoring research [33,34,35]. The MODIS NDVI used in this study was acquired from the MODIS 13Q1 Vegetation Index data set product (Geospatial Data Cloud Platform of the Computer Network Information Center of the Chinese Academy of Sciences,http://www.gscloud.cn/ (accessed on 10 January 2022)), with a spatial resolution of 500 m × 500 m and the temporal resolution of 30 days. To smooth the period of NDVI, the self-adapted filtering operation Savitzky–Golay was used to reduce the influences caused by noise such as outliers. GIMMS NDVI3g data set was downloaded from the ECOCAST website (https://ecocast.arc.nasa.gov/data/pub/gimms/3g.v1/ (accessed on 12 January 2022)), with the spatial resolution of 8 km × 8 km and the temporal resolution of 15 days.

These two kinds of NDVI original data sets have discrepancies in spatial and temporal resolutions. In order to enhance the fusion of the two data sets, reduce the influence of cloud and other atmospheric effects on the data, and improve the data accuracy our study adopted the maximum synthesis method to synthesize the MODIS NDVI data sets in 2000, 2005, 2010, and 2015 and GIMMS NDVI data sets in 1990, 1995, 2000, 2005, 2010 and 2015, respectively [36]. In addition, to meet the data requirements of the ESTARFM, the spatial resolution of the GIMMS NDVI data sets was configured to 500 m × 500 m after resampling, using the nearest neighbor pixel method.

#### 2.2.2. Climatic and Auxiliary Data

The meteorological data set includes the average annual temperature and annual precipitation in 1990, 1995, 2000, 2005, 2010, and 2015, with a spatial resolution of 1 km × 1 km and the terrain data spatial resolution of 500 m × 500 m. The climate data set, topographic data set, and Qinghai–Tibet Plateau partitioned data were obtained from the Resource and Environmental Science Data Center of the Chinese Academy of Sciences (http://www.resdc.cn (accessed on 20 January 2022)). The vegetation type data came from the National Tibetan Plateau Data Center (https://www.tpdc.ac.cn (accessed on 3 February 2022)). The climate data set were adjusted with Australia’s ANUSPLIN interpolation software to interpolate climatic factors such as temperature and precipitation. Specific principles are provided in the reference [37].

To unify the data standards, the spatial resolution of the data set resampled by the nearest-neighbor pixel method is 500 × 500 m. All data were converted into a unified projected coordinate system WGS_1984_Zone_48N and cropped out in the study area.

### 2.3. Methods

#### 2.3.1. ESTARFM Spatiotemporal Fusion Algorithm

Obtained the ESTARFM algorithm by improving the STARFM algorithm [28]. This algorithm can generate fused data with high time and space resolution by combining the time or space advantages of the two types of data sources. Based on the original model, the weighting method was adjusted by the ESTARFM fusion model according to the heterogeneity of the pixels, thus the prediction results by setting conversion coefficients were improved (Algorithm 1). Greater heterogeneity led to higher prediction accuracy, which preserved more details of spatial features [4]. In this study, based on MODIS NDVI with a spatial resolution of 500 m × 500 m, the ESTARFM algorithm was used to obtain the fused GIMMS NDVI3g data from 1990 and 1995 in the Tibetan Plateau, respectively, with a spatial resolution of 500 m × 500 m. As shown in Figure 1:
**Algorithm 1:** Pseudocode of the ESTARFMInput: two fine-resolution images at tm and tn, three coarse-resolution images at tm , tn and tpOutput: fine-resolution image at tp1: If F(x,y,tk,B)=a×C(x,y,tk,B)+b :2: Tk=1/|∑j=1w∑l=1wC(xj,yl,tk,B)−∑i=1w∑l=1wC(xj,yl,tp,B)|∑k=m,n(1/|∑j=1w∑l=1wC(xj,yl,tk,B)−∑i=1w∑l=1wC(xj,yl,tp,B)|) ,(k=m,n).3: Then4: F(xw/2,yw/2,tp,B)=Tm×Fm(xw/2,yw/2,tp,B)+Tn×Fn(xw/2,yw/2,tp,B)5: Check convergence6: |F(xi,yi,tk,B)−F(xw/2,yw/2,tk,B)|≤δ(B)×2/m7: Compute average absolute difference (AAD) and average absolute (AD)

After preprocessing such as registration and cropping of the two images, it is assumed that the systematic deviation between MODIS NDVI and GIMMS NDVI was solely attributed to the discrepancy in NDVI values, and there was no significant difference between the images in the two periods, as shown in Formula (1).
(1)$$M\left(x,y,t_{p},B\right)=M\left(x,y,t_{0},B\right)+a\times \left(G\left(x,y,t_{p},B\right)-G\left(x,y,t_{0},B\right)\right)$$

In which M represents the MODIS NDVI data, G represents the GIMMS NDVI3g data, (x,y) represents the pixel location, B represents the image band, $t_{0}\;$and $t_{p}\;$represent the time of data acquisition, and a is the conversion coefficient, depending on the system error of the sensor.

Complexity and uncertainty of the actual surface conditions caused deviations in the actual pixel information prediction, therefore it was assumed that the change of the NDVI value of the mixed pixel is a typical epitome of NDVI amongst different land cover types, and the change of the mixed pixel NDVI over time is the weighted sum of the NDVI variations of the pixels from different land cover types. The proportion of each component in different land cover types remains unchanged, as shown in Formula (2).
(2)$$M\left(x,y,t_{p},B\right)=M\left(x,y,t_{0},B\right)+v\left(x,y\right)\times \left(G\left(x,y,t_{p},B\right)-G\left(x,y,t_{0},B\right)\right)$$

v(x,y)  represents the correlation coefficient corresponding to the i-th similar pixel after the decomposition of mixed pixels. Formula (2) reveals that the change in the NDVI value of a pixel is the most similar to its neighbors. Images were fused according to the following steps. Using the moving window set by the adjacent pixels, similar pixels can be found. Whilst using the mutual relationship among the pixels, the value of the center pixel of the image can be obtained, as shown in Formula (3).
(3)$$M\left(x_{w/2},y_{w/2},t_{p},B\right)=M\left(x_{w/2},y_{w/2},t_{0},B\right)+\sum_{i=1}^{N}W_{i}\times v_{i}\times \left(G\left(x_{i},y_{i},t_{p},B\right)-G\left(x_{i},y_{i},t_{0},B\right)\right)$$

In which N represents the number of similar pixels in the center prediction pixel, $\left(x_{i},y_{i}\right)$, W, and $v_{i}$ represent the position, the weight, and the conversion coefficient of the i-th similar pixel, respectively.

The MODIS NDVI image at a time $t_{p}\;$is obtained by fusing the MODIS NDVI at time m in the first period and the GIMMS NDVI3g at a time $t_{p}$, which is recorded as $M_{m}\;\left(x_{w/2},y_{w/2},t_{p},B\right)$. The observation data at a time n in the second period and the MODIS NDVI data at the time $t_{p}$ were fused and denoted as $M_{n}\;\left(x_{w/2},y_{w/2},t_{p},B\right)$. The MODIS NDVI data at the time $\;t_{p}$ tended to be more accurate after weighting the two results. The weights are calculated by the changes between GIMMS at the time $t_{m}$ and time $t_{n}$, and the GIMMS at time $t_{p}$, respectively, as shown in Formula (4).
(4)$$T_{k}=\frac{1/\left|\sum_{j=1}^{W}\sum_{i=1}^{W}G\left(x_{i},y_{j},t_{k},B\right)-\sum_{j=1}^{W}\sum_{i=1}^{W}G\left(x_{i},y_{j},t_{p},B\right)\right|}{\sum_{k=m,n}\left(1/\left|\sum_{j=1}^{W}\sum_{i=1}^{W}G\left(x_{i},y_{j},t_{k},B\right)-\sum_{j=1}^{W}\sum_{i=1}^{W}G\left(x_{i},y_{j},t_{p},B\right)\right|\right)},\left(k=m,n\right)$$

The value of the center pixel can be obtained by Formula (5):(5)$$M\left(x_{w/2},y_{w/2},t_{p},B\right)=T_{m}\times M_{m}\;\left(x_{w/2},y_{w/2},t_{p},B\right)+T_{n}\times M_{n}\;\left(x_{w/2},y_{w/2},t_{p},B\right)$$

#### 2.3.2. The Calculation of Vegetation Coverage

Vegetation coverage, as an important indicator for studying global climate change and vegetation growth status, has received extensive attention from scholars worldwide [33,35,38]. The calculation method is shown in Formula (6):(6)$$F=\left(NDVI-NDVI_{n}\right)/\left(NDVI_{i}-NDVI_{n}\right)$$
where F  represents the vegetation coverage; NDVI  is the normalized vegetation index of the pixel; $NDVI_{i\;}$ and $NDVI_{n\;}$ are, respectively, the vegetation index of vegetation-covered land and bare soil land. NDVI values with 95% confidence $\left(NDVI_{i\;}\right)$ and 5% confidence $\;(NDVI_{n\;})\;$ in the study area were chosen.

#### 2.3.3. Linear Regression Analysis

The simple linear regression model was used to assist the calculation of various characteristics and laws of vegetation coverage in single-pixel [39].
(7)$$A=\frac{m\times \sum_{i=1}^{m}\left(i\times F_{i}\right)-\left(\sum_{i=1}^{m}i\right)\times \left(\sum_{i=1}^{m}F_{i}\right)}{\left(m\times \sum_{i=1}^{m}i^{2}-\sum_{i=1}^{m}i^{2}\right)}$$
where A  represents the change rate of vegetation coverage over time in the research period; m represents the total research years; and $F_{i\;}$ represents the vegetation coverage of the i-th year. Thus, the variation range of vegetation coverage can be calculated as follows:(8)B=A×(n−1)

If ***A*** value is negative, the vegetation coverage of pixels from 1990 to 2015 shows A downward trend; otherwise, it shows an upward trend. If A value is zero, it shows no obvious change in trend; the larger the absolute value of A is, the more obvious the change is.

#### 2.3.4. Hurst Index

According to the Hurst index, based on the R/S analysis method, the continuity of vegetation coverage over time can be calculated effectively, as long as the value is in the range of [0, 1]. It has received extensive attention in hydrology [40,41], geography [42], climatology [43], and so on. The calculation procedure is demonstrated as follows:

The mean of vegetation coverage over time is determined by Formula (9), with the time set t=1,2,…,n.
(9)$$\bar{f}\left(\tau \right)=\frac{1}{\tau }\sum_{t=1}^{\tau }f\left(t\right),\tau =1,2,\ldots ,n$$

The cumulative deviation of time t:(10)$$X\left(t,\tau \right)=\sum_{t=1}^{\tau }\left(f\left(t\right)-\bar{f}\left(t\right)\right),1\leq t\leq \tau$$

The range of vegetation coverage over time set:(11)$$R\left(\tau \right)=\underset{1\leq t\leq \tau }{\max}X\left(t,\tau \right)-\underset{\;1\leq t\leq \tau }{\min}X\left(t,\tau \right),\;\tau =1,2,\ldots ,n$$

Construct the standard deviation sequence:(12)$$S\left(\tau \right)=\sqrt{\frac{1}{\tau }\sum_{t=1}^{\tau }{\left(f\left(t\right)-\bar{f}\left(t\right)\right)}^{2}},\;\tau =1,2,\ldots ,n$$

Hurst index:(13)$$R\left(\tau \right)/S\left(\tau \right)={\left(c\tau \right)}^{H}$$

In the formulas above, H is the Hurst index. Time set t is not continuous with H=0.5. Time set t has a distinctive feature of continuity with H>0.5 . The continuous trend becomes more and more obvious as H is approaching 1. The time set has a distinctive feature of anti-continuity when H<0.5 . The closer H is to 0, the more obvious the anti-continuous trend is.

#### 2.3.5. Partial Correlation Analysis

The complexity and the correlation which exist between elements of geographic systems render the relation between driving factors become the key to driving analysis. Using partial correlation analysis, the correlation between vegetation coverage and a single, climate-driving factor can be determined by ruling out the influences of other climate driving factors [44].
(14)$$R_{ab,c}=\frac{R_{ab}-R_{ac}R_{bc}}{\sqrt{\left(1-R_{ac}{}^{2}\right)+\left(1-R_{bc}{}^{2}\right)}}$$

In the formula, $R_{ab,c}$ is the partial correlation index between a and b, based on the control variable c ; $R_{ab}$, $R_{bc\;}$, and $R_{ac}$ are, respectively, the simple correlation coefficients between a and b, b  and c, and a and c, in which a, b, and c are, respectively, the vegetation coverage index, mean annual temperature, and annual precipitation. Two variables are positively correlated, when $\;R_{ab,c}>0$, and vice versa. Greater absolute value $R_{ab,c}\;$ stands for a greater correlation between the two variables.

The calculation of the simple correlation coefficient is as follows:(15)$$R_{xy}=\frac{\sum_{i=1}^{n}\left(\left(x_{i\;}-\bar{x}\right)\left(y_{i\;}-\bar{y}\right)\right)}{\sqrt{\sum_{i=1}^{n}{\left(x_{i\;}-\bar{x}\right)}^{2}\sum_{i=1}^{n}{\left(y_{i\;}-\bar{y}\right)}^{2}}}$$
where, $R_{xy\;}\;$ is the simple correlation coefficient between x and y, with the range of [−1, 1]; $x_{i\;}\;$ is the value of the variable x in the i-th year; $y_{i\;}\;$ is the value of a variable y in the i-th year; $\bar{x}\;$ is the mean of a variable x during the study period; $\bar{y}\;$ is the mean of a variable y during the study period.

## 3. Results

### 3.1. Data Fusion

We summarized and plotted the values of pixels in the fused GIMMS NDVI3g image and the real MODIS NDVI image in 2000, 2005, 2010, and 2015, as shown in Figure 2. The fused GIMMS NDVI3g data and the real MODIS NDVI data were in good agreement. The scattered points of the pixels were evenly distributed along the trend line of y=x, indicating a decent prediction effect.

By comparing the details of GIMMS NDVI3g data, fused GIMMS NDVI3g data, and real MODIS NDVI data in 2000, 2005, 2010, and 2015 (Figure 3), higher pixel resolutions were observed with the fused GIMMS NDVI3g data and real MODIS NDVI data, where the texture of the ground features was clear and the details of some characteristic areas can be identified. In addition, the detailed features of the regional vegetation coverage can be well reproduced.

### 3.2. Analysis of Vegetation Coverage Characteristics

The Qinghai–Tibet Plateau is a vast territory with complex and diverse climate types. In order to better analyze the vegetation distribution and change characteristics of the Qinghai–Tibet Plateau, the study area was divided into 18 zones according to the Qinghai–Tibet Plateau zoning data and vegetation coverage type data. (Table 1). Notably, the VIIBiib, VIIIBi, VIIICi, VIIICii, and VIIIBii zones had more obvious vegetation degradation trends, whereas the VIAiic zone had a more obvious vegetation improvement trend. The absolute values of A in all the zones were above 0.8%, implying that the vegetation changes in the Qinghai–Tibet Plateau had extreme distribution characteristics. In areas with low average vegetation coverage, such as desert zones and sub-zones, the vegetation tends to degenerate, while vegetation trends improve as the average vegetation coverage becomes higher.

In 2000, the State Council issued the “Several Opinions on Doing a Good Job in Returning Cultivated Land to Forest and Grassland” [45]. Returning farmland to forest and grassland policy was implemented nationwide in China, which generally promoted vegetation growth. In order to explore the change in vegetation on the Qinghai–Tibet Plateau since the promulgation of this policy, the profile of vegetation coverage and Hurst index of the Qinghai–Tibet Plateau around 2000 and 1990–2015 were calculated in this study, as shown in Figure 4. From 1990 to 1995, 37.46% of the area was subject to vegetation degradation. The land with A values less than −0.5%, between −0.5% and 0%, and between 0% and 0.5% accounted for 9.93%, 27.53%, and 54.42% of the total area, respectively, while the land with A values over 0.5% accounted for 8.12% of the total area. From 2000 to 2015, 52.47% of the area suffered from vegetation degradation, with an average growth trend (A) of 0.02%. The land with A values less than −5%, between −5% and 0%, and between 0% and 5% accounted for 0.23%, 52.24%, and 47.37% of the total area, respectively, while the land with A values over 5% accounted for 0.15% of the total area. From 1990 to 2015, the area with vegetation degradation increased to 73.49% with an average growth trend (A) of −0.041%. The land with A values less than −5%, between −5% and 0%, between 0% and 5%, and over 5% accounted for 0.10%, 73.39%, 26.42%, and 0.08% of the total area, respectively. According to the distribution of vegetation temporal profile from 1990 to 2015 (Figure 4c), it is clear that the vegetation changes on the Qinghai–Tibet Plateau were spatially heterogeneous and specifically manifest as degradation in the west and rise in the east. Furthermore, the degradation of vegetation was prevalent in the study area. Areas with a Hurst index less than 0.5, over 0.5, and equal to 0.5 accounted for 9.16%, 80.59%, and 10.25% of the total area, respectively. The spatial distribution of the Hurst index indicated that the change of vegetation characteristics in most parts of the Qinghai–Tibet Plateau was continuous, whereas the continuity was not evident in the western Sichuan and the northern Yunnan regions. The comprehensive vegetation change trend and Hurst index showed that 22.14% of the Tibetan Plateau continued to improve from 1990 to 2015 (A>0 and H>0.5).

### 3.3. Impact of the Topographic on Vegetation Coverage

The terrain of the Qinghai–Tibet Plateau is complex and changeable, with ascending attitude from the southeast toward the northwest. Regarding previous studies, this paper extracted two types of terrain factors, namely slope, and aspect, to explore their driving mechanism for vegetation change [46]. The slope was divided into 0–3, 3–8, 8–15, 15–25, and greater than 25 degrees by the natural breakpoint method. At the same time, the slope aspect was calculated by the elevation data, and the aspect data was divided into three categories: flat slopes with a result of −1; sunny slopes were generally south, southwest, west, and northwest and the value range was between 157.5°and 337.5°; shady slopes were generally northeast, east, north, and southeast and the value range was 337.5°–360° and 0°–157.5°. In the aspect of the spatial scale, when the slope was over 15°or located on the slope surface, the vegetation cover was in good condition. In terms of the time scale, the vegetation coverage driven by terrain factors was developing toward continuous degradation (A<0 and Hurst > 0.5). The degradation trend appeared to be less significant with steeper slopes. It should be highlighted that when the terrain slope was between 0°and 3°, the vegetation coverage in 1990 and 2015 was 0.38 and 0.36, respectively. In contrast, with the A value of -0.58% (Hurst > 0.6) and the terrain slope of over 25°, the vegetation coverage in 1990 and 2015 increased to 0.71 and 0.70, respectively. In 2015, the vegetation coverage was 0.71 and 0.70, respectively. Moreover, as the A value came to −0.08% (Hurst > 0.6), the impact of this aspect on the vegetation coverage was smaller than that of the slope (Table 2).

### 3.4. Link between Climate Change and Vegetation Coverage

Based on the partial correlation analysis (Figure 5), it could be found that the average annual temperature and annual precipitation in different areas of the Qinghai–Tibet Plateau from 1990 to 2015 had significant differences due to the interference of vegetation coverage. In general, their changes drove the decrease in vegetation coverage, with annual mean temperature having a greater impact on vegetation coverage than annual precipitation. Rt > 0, that is, 9.59% of the total area was driven by the change of average annual temperature promoting vegetation coverage. A lot of the 47.67% of the land had Rp > 0, where the change of annual precipitation promotes vegetation coverage, while the change in annual precipitation was negatively correlated with the vegetation coverage in the rest of the area.

In the alpine region of the Qinghai–Tibet Plateau, vegetation coverage was positively correlated with annual mean air temperature and negatively correlated with annual precipitation in the alpine shrub and meadow zone (VIIIAi) and in the alpine desert zone (VIIICi). The annual precipitation and the average annual temperature, respectively, had a negative and positive correlation with the vegetation coverage. In the temperate desert region, the average annual temperature and annual precipitation had significant differences in the driving degree of vegetation coverage changes, which showed that the correlation degree between vegetation coverage and the annual air temperature was greater than that of annual precipitation in the temperate shrub/semi-shrub desert sub-region (VIIBib) and alpine desert region (VIIICi).

The degree of correlation is greater than the annual precipitation. The average annual temperature had a stronger correlation with the vegetation coverage compared to the annual precipitation.

The changes in vegetation coverage, temperature, and precipitation with time are shown in Figure 6. In general, global warming caused by the increase of annual average temperature had a positive effect on the alpine vegetation change on the Qinghai–Tibet Plateau but a negative effect on the vegetation change in the temperate desert regions. The increase in annual precipitation had a significant negative effect on alpine vegetation and temperate desert vegetation on the Tibetan Plateau, presumably due to the hysteresis, with a certain hysteresis effect. The annual vegetation coverage changes in the alpine shrub and meadow zone (VIIIAi) and alpine desert zone (VIIICi) were more sensitive to the average annual temperature than to the annual precipitation, due to the location of the alpine region. Vegetation coverage had a significant positive correlation with the average annual temperature and a weakly negative correlation with the annual precipitation. In addition, hysteresis was one of the driving factors influencing the annual precipitation (Figure 6a–d). Although both belong to the temperate desert region, the temperate shrub/semi-fruticous desert zone (VIIBib) and the temperate semi-shrub/fruticous desert zone (VIIBi) showed that the vegetation coverage was affected by the average annual temperature due to different water and heat conditions. There were differences in the driving degree and driving direction of annual precipitation. In temperate shrub and semi-fruticous desert zone (VIIBib), the average annual temperature had a positive effect on vegetation coverage, while the annual precipitation had a negative driving effect. Furthermore, the impact of annual precipitation before 2010 was more obvious. The combined effect of annual average temperature and annual precipitation was responsible for the improvement of vegetation coverage after 2010. In the temperate semi-shrub and fruticous desert zone (VIIBi), the average annual temperature and annual precipitation, with similar growth curves, both showed a negative effect on vegetation coverage (Figure 6e–h).

## 4. Discussion

Biological and environmental factors were two main causes for the change of NDVI, and the latter had a stronger influence [47]. Therefore, it is reasonable for the ESTARFM model to assume that the change characteristics in a small range are similar [28]. The adjustment of weighting and conversion coefficient can improve the results to a certain extent and retain more spatial details. The high spatiotemporal-resolution image that fused in this study, based on the ESTARFM, had preeminent spatial details and a strong correlation with the original MODIS data (R^2^ ≥ 0.95, *p* < 0.01), which facilitated the study of the characteristics of vegetation changes on the Qinghai–Tibet Plateau. This result demonstrated the feasibility of the ESTARFM model to study the vegetation change characteristics of the Tibetan Plateau, and further proved the scientific nature of the ESTARFM model to integrate medium- and high-resolution images.

From 1990 to 2015, the vegetation coverage of different slopes and slope directions on the Qinghai–Tibet Plateau showed a trend of degradation. When the slope was more than 25°, the vegetation coverage had the weakest degradation trend (A=−0.08%, Hurst = 0.64), and the slope with the strongest degradation trend was between 0°and 3° (A=−0.58%, Hurst = 0.67). According to the analysis, the policy implemented by the State Council in 2000, that all the farmland with a slope of >25°should be returned to forests and grasslands in the Qinghai–Tibet Plateau area, was responsible for the improved vegetation coverage. The areas with a slope between 0°and 3° were mainly intermountain basins and low mountains, where the degradation of grassland was significant and widespread, and the ecological environment was poor. The data suggest that restoration measures, such as returning farmland, further protection, and restoration need to be carried out in order to achieve significant results. The vegetation cover and its degradation degree on the shady slope was slightly higher than that of the sunny slope (Table 2), which further proves that the vegetation cover change on a shady slope was more sensitive than that of the sunny slope [46].

The total change percentage for vegetation coverage was −0.41% from 1990 to 2015 and 0.02% from 2000 to 2015. The results showed that the conversion of cropland to forest and grassland in China since 2000 improved the vegetation coverage in this region to a certain extent, which further proved that the vegetation coverage in the Tibetan Plateau had decreased [48].

From 1990 to 2015, the vegetation coverage on the Tibetan Plateau increased in the east while decreasing in the west with the larger land area (Figure 4c). In 2010, the vegetation coverage of the Qinghai–Tibet Plateau changed significantly as well as the average annual temperature and the annual precipitation (Figure 6). Previous studies and analyses illustrated that the spatiotemporal distribution of major climatic factors is one of the main reasons for their complexity [49,50]. The analysis of this study showed that the distribution, trend, and direction of vegetation coverage were affected differently by temperature and precipitation due to regional differences in different zones. The northeast and northwest of the Qinghai–Tibet Plateau were mainly deserts, where vegetation coverage had a strong and positive correlation with temperature but a negative correlation with precipitation. The data indicated that vegetation coverage was sensitive to changes in water and heat conditions. In this study, it was found that the precipitation change in the northeast was the main reason for vegetation degradation (Figure 6), which was consistent with previous research results [5,51]. In addition, a stronger positive correlation between temperature and vegetation coverage in the alpine desert area of the Qinghai–Tibet Plateau was observed, and the change of hysteresis in vegetation coverage caused by precipitation is more significant compared to temperature. On the southeastern Qinghai–Tibet Plateau, temperature and precipitation had similar change trends, and they both negatively affected vegetation coverage. 

The area was mainly located in temperate grassland and subtropical broad-leaved forest, with good water and heat conditions. Therefore, in terms of climate impact, it was believed that the effects of temperature and climate on vegetation growth in the area had reached equilibrium or saturation. If significant climate change occurs, the growth of vegetation will be inhibited. Combining our results with those of previous studies [5], it was concluded that the uncertainty of other factors was an important reason for the de-concentration of partial correlation between vegetation coverage and temperature and precipitation in the northwest part of the Qinghai–Tibet Plateau (Figure 5). Therefore, further studies are needed to determine the driving factors of vegetation change in this area.

The ESTARFM model required two pairs of input data, and the accuracy of the input data was high; due to the high altitude of the Qinghai–Tibet Plateau, the cloud cover was more serious, which affected the accuracy of data fusion. The spatial resolution of the data fusion in this study was 500 m. Therefore, follow-up research can calculate the vegetation coverage before fusion and then remove and process outliers. The influencing factors of vegetation change are complex, and future research should consider this more comprehensively and use higher spatial resolution data to study the vegetation conditions of the Qinghai–Tibet Plateau.

## 5. Conclusions

The fused NDVI results based on the ESTARFM model were used to study vegetation changes on the Tibetan Plateau with complex terrain. From 1990 to 2015, the overall vegetation coverage of the Qinghai–Tibet Plateau was degraded by 0.41%. However, under the policy of returning farmland to forest and grassland, the overall degradation trend degree and the degradation trend degree of slope ≥ 25° were weakened. In a local scale, the area of vegetation reduced in the west was greater than that of vegetation increased in the east. The regional variances in different zones led to various influences on temperature and precipitation, in terms of distribution, change degree, and change in the direction of vegetation coverage. The main causes of the significant changes in 2010 were attributed to the spatial distribution of climate factors such as temperature and precipitation. This study evaluated the vegetation coverage of the Qinghai–Tibet Plateau and revealed its key characteristics for terrain and some climatic factors. The knowledge generated from this study provides an important scientific basis for local government decision-making and economic development.

The data used in this study has a long time interval, and some details may be neglected. Two main climatic factors, average annual temperature and annual precipitation, were selected for this study. Given the complexity of the geographic system and human activities, other types of climatic factors should be considered as well. Thus, the specific response mechanism needs to be further studied.

## Figures and Tables

**Figure 1 ijerph-19-08836-f001:**
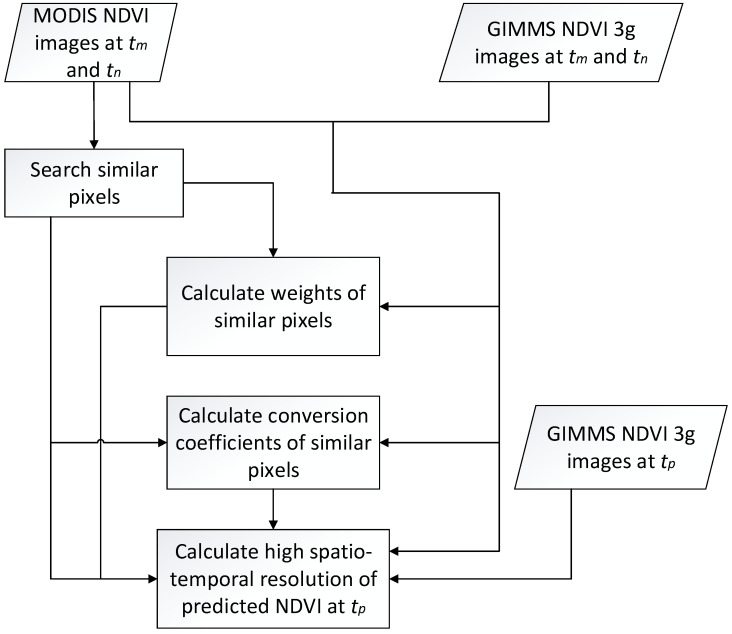
Schematic explanation of the enhanced spatial and temporal adaptive reflectance fusion model (ESTARFM) for fusing GIMMS NDVI3g and MODIS NDVI.

**Figure 2 ijerph-19-08836-f002:**
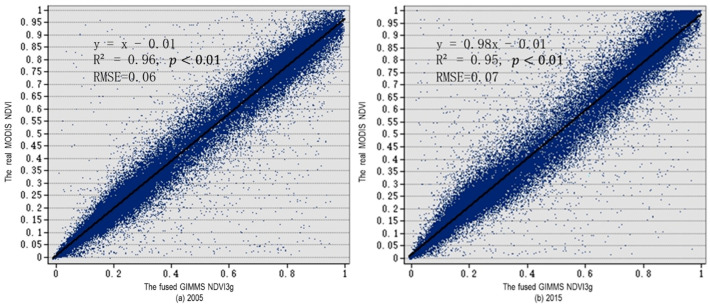
Scatter plot of pixel values corresponding to the fused GIMMS NDVI3g and MODIS NDVI.

**Figure 3 ijerph-19-08836-f003:**
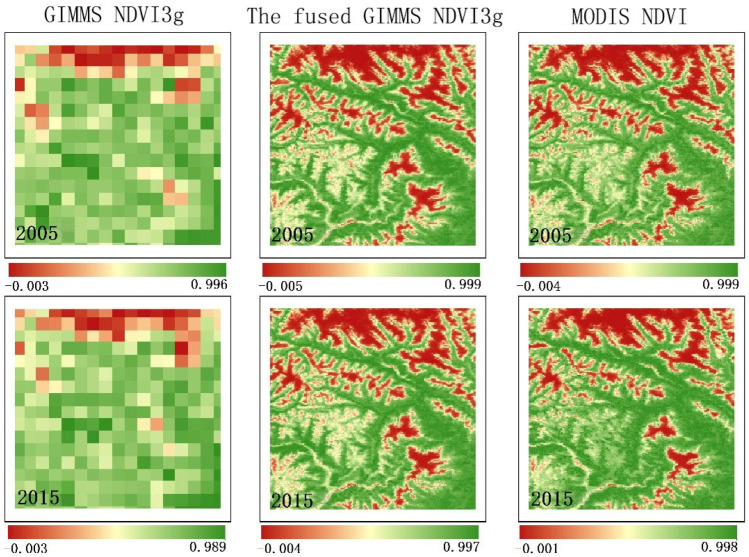
Detail comparison of fused GIMMS NDVI3g and real MODIS NDVI data.

**Figure 4 ijerph-19-08836-f004:**
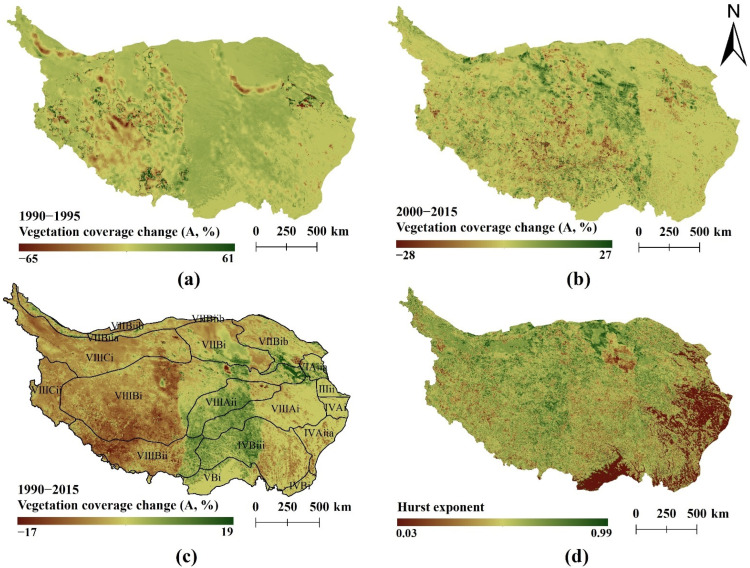
The vegetation change trend and Hurst index on the Qinghai–Tibet Plateau, (**a**) is the vegetation change trend from 1990 to 1995, (**b**) is the vegetation change trend from 2000 to 2015, (**c**) is the vegetation change trend from 1990 to 2015, (**d**) is the Hurst index from 1990 to 2015.

**Figure 5 ijerph-19-08836-f005:**
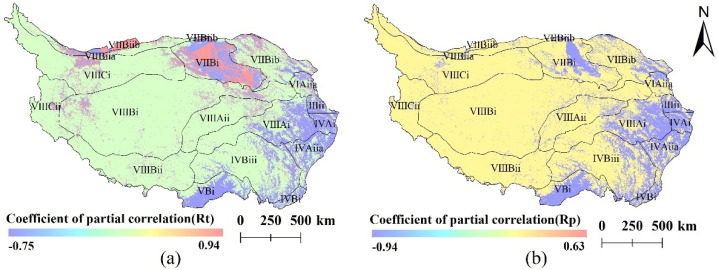
The coefficient of partial correlation between vegetation coverage and climate factors, (**a**) is the coefficient of partial correlation between vegetation coverage and average annual temperature, and (**b**) is the coefficient of partial correlation between vegetation coverage and annual precipitation.

**Figure 6 ijerph-19-08836-f006:**
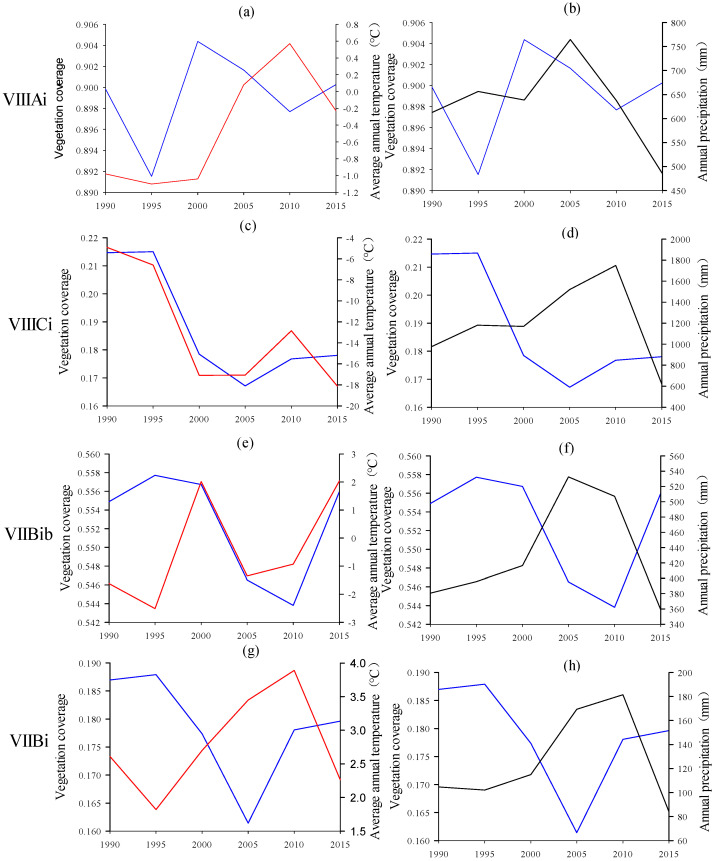
The change curves of vegetation coverage, average annual temperature, and annual precipitation in VIIIAi (**a**,**b**), VIIICi (**c**,**d**), VIIBib (**e**,**f**), VIIBi (**g**,**h**). The change curves of vegetation coverage, average annual temperature, and annual precipitation are, respectively, blue, red, and black.

**Table 1 ijerph-19-08836-t001:** Characteristics of different zones on the Qinghai–Tibet Plateau.

Zones	Codes	Average Vegetation Coverage		A Value(%) of 1990–2015
		1990	2015	
Warm temperate shrubbery, semi-shrubbery, and bare land zone	VIIBiib	0.17	0.13	−0.87
Southern temperate sylvosteppe zone	VIAiia	0.84	0.86	0.45
Southern temperate desert steppe sub-zone	VIAiic	0.75	0.8	1.03
Alpine steppe zone	VIIIBi	0.39	0.36	−0.84
Alpine shrub and meadow zone	VIIIAi	0.89	0.9	0.19
Alpine meadow zone	VIIIAii	0.68	0.72	0.58
Cold temperate coniferous forest zone in subtropical mountain	IVBiii	0.8	0.81	0.2
Warm temperate shrub, semi-shrub and desert sub−zone	VIIBiia	0.3	0.29	−0.38
Alpine desert zone	VIIICi	0.21	0.18	−0.88
Temperate desert zone	VIIICii	0.3	0.25	−1.08
Temperate steppe zone	VIIIBii	0.53	0.49	−0.92
Mid-subtropical evergreen broad-leaved forest zone	IVBi	0.99	0.98	−0.24
North tropical seasonal rain forest, semi evergreen season	VBi	0.87	0.87	0.1
Northern mid-subtropical evergreen broad-leaved forest sub-zone	IVAiia	0.95	0.94	−0.35
Northern subtropical mixed evergreen and deciduous broad-leaved zone	IVAi	0.98	0.98	−0.07
Warm temperate deciduous oak forest zone	IIIii	0.98	0.98	0.03
Temperate shrub and semi-fruticous desert zone	VIIBib	0.55	0.56	−0.13
Temperate semi-shrub and fruticous desert zone	VIIBi	0.19	0.18	−0.23

**Table 2 ijerph-19-08836-t002:** Vegetation coverage and its variation characteristics on the Qinghai–Tibet Plateau under different terrain factors.

Topographic Factors		Average Vegetation Coverage		A Value (%) from 1990 to 2015	Hurst Index from 1990 to 2015
		1990	2015		
Slope (°)	0–3	0.38	0.36	−0.58	0.67
	3–8	0.5	0.5	−0.51	0.67
	8–15	0.58	0.57	−0.37	0.65
	12–25	0.63	0.62	−0.23	0.65
	≥25	0.71	0.7	−0.08	0.64
Aspect	Flat	0.35	0.34	−0.45	0.65
	Sunny slope	0.52	0.5	−0.4	0.66
	Shady slope	0.53	0.51	−0.44	0.66

## Data Availability

The data presented in this study are available contained within the article.

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
