# Peer review of "Study on Spatiotemporal Variation Pattern of Vegetation Coverage on Qinghai–Tibet Plateau and the Analysis of Its Climate Driving Factors"

_ijerph, 2022, doi:10.3390/ijerph19148836_

Round 1

Reviewer 1 Report

In this review, I omitted a few minor comments regarding the manuscript editing. It presents only the most important allegations which are the basis for my negative assessment.

  1. There are many studies on the analysis of vegetation driving factors in the Qinghai-Tibet Plateau based on large-scale and high-resolution dataas follows:

Diao C, Liu Y, Zhao L, et al. Regional-scale vegetation-climate interactions on the Qinghai-Tibet Plateau[J]. Ecological Informatics, 2021, 65:101413.

Duan H, Xue X, Wang T, Kang W, Liao J, Liu S. Spatial and Temporal Differences in Alpine Meadow, Alpine Steppe and All Vegetation of the Qinghai-Tibetan Plateau and Their Responses to Climate Change. Remote Sensing. 2021; 13(4):669.

Wang H, Qi Y, Huang C, et al. Analysis of vegetation changes and dominant factors on the Qinghai-Tibet Plateau, China[J]. Sciences in Cold and Arid Regions, 2019, 11(2):62-70.

Therefore, it seems there are no innovations in this study.

  1. 500 m NDVI is not high-resolution data.

  1. Downscaling the remotely sensed vegetation index datasets with ESTARFM is not an innovation either.

  1. It’s confusing that how did you divided the study area into 18 zones and why.

Author Response

Thank you for processing our submission and giving us the opportunity to revise.

In response to the problem of resolution, we rechecked the articles on the vegetation cover of the Qinghai-Tibet Plateau, including the three proposed literatures, and found that most of the relevant studies were based on the time period after 2000. They were basically 500m and 1000m, and some had longer research years and used GIMMS data with lower spatial resolution. This paper was aimed at this problem, extending the research time and improving the spatial resolution. Subsequent research may consider fusion with Landsat data to further improve the spatial resolution of the data source.

For the problem that the research area was divided into 18 areas: based on the vegetation type data mentioned in the data source, the research area was divided into 18 areas.

Reviewer 2 Report

A novel study on the analysis of vegetation driving factors in the Qinghai-Tibet Plateau using high spatial resolution data and appliyng the Enhanced Spatial and Temporal Adaptive Reflectance Fusion model is proposed.

Authors must clearly highlight the limitations of recent studies in the literature and how their approach intends to overcome them. In particular, it is necessary to highlight the performance advantages of the proposed approach.

Paragraph 2.3 needs to be revised in a more structured fashion. A figure is missing that shows the architectural outline of the proposed method and it is necessary to outline the processes used by adopting a sequential order.

A pseudocode structuring of the ESTARFM algorithm is required. It is recommended to quote the formulas used numerically in the algorithm.

A brief description of the GIS-based and geostatistical environments used in the experimental test should be inserted before showing the results.

The classifications of Slope and Aspect in Tab. 2 must be justified. In particular, it is necessary to specify which are the sources of these classifications and, for the three classes of Aspect, to which ranges in [0,255] the three classes of Aspect Flat, Sunny Slope and Shady Slope refer.

What are the future research developments? What are its performance limits and how could they be overcome? Authors should include a discussion of these points in the concluding section.

Author Response

In response to the limitation of recent literature research, the second and third paragraphs of the introduction were revised to briefly summarize the current research status and limitations of the literature.

The second and third paragraphs of the article have been re-edited, and the process of the entire article was described in the last paragraph of the introduction. Added Figure 1 to the article, describing the implementation flow of the ESTARFM model.

The implementation formula of the ESTARFM model is described in the article, and the corresponding reference articles were cited. At the same time, when the reworked article was submitted, the relevant code files were attached.

For my understanding of the fourth question, I have added a relevant description in the overview of the study area.

Table 2 provided relevant descriptions for the classification of slope and aspect, in which the slope was divided into 5 categories according to the natural breakpoint method; the aspect was divided into three categories according to the results of the calculation of the elevation data.

For the progress and limitations of future research, a description was added in the discussion section.

Round 2

Reviewer 1 Report

I appreciate the effort from the authors in providing additional material in response to my comments. I think the manuscript could be accepted now.

Author Response

Thank you for processing our submission and for your valuable comments to improve the quality of the manuscript. Also, thank you very much for agreeing to accept the manuscript

Reviewer 2 Report

Authors took my suggestions into account, improving the quality of the manuscript. Still missing a structured description in pseudocode mode of the ESTARFM algorithm; It must be added in the description of the methodology adopted, regardless of the presence of any attached code.

Author Response

Thank you for working on our revised manuscript and for giving us a second chance to revise.

Based on your comments, we have added pseudocode for the ESTARFM algorithm in the Methods.